# Food Selectivity and Its Implications Associated with Gastrointestinal Disorders in Children with Autism Spectrum Disorders

**DOI:** 10.3390/nu14132660

**Published:** 2022-06-27

**Authors:** Angel F. Valenzuela-Zamora, David G. Ramírez-Valenzuela, Arnulfo Ramos-Jiménez

**Affiliations:** Department of Chemical Biological Sciences, Institute of Biomedical Sciences, Autonomous University of Ciudad Juárez, Ciudad Juárez, Chihuahua 32310, Mexico; angel.valenzuela@uacj.mx (A.F.V.-Z.); david.ramirez@uacj.mx (D.G.R.-V.)

**Keywords:** neurobiological affectations, developmental disorder, food neophobia, autism, constipation, diarrhea, oral over-responsiveness

## Abstract

Food selectivity (FS) in children with autism spectrum disorders (ASD) is common, and its impact on a nutritional level is known. However, the etiology of gastrointestinal disorders (GID) related to alterations in the intestinal microbiota in children with ASD remains unclear. This article provides a narrative review of the literature on FS from the last 15 years, and its relationship with GID in children with ASD. Sensory aversion in ASD leads to food elimination, based on consistencies, preferences, and other sensory issues. The restriction of food groups that modulate the gut microbiota, such as fruits and vegetables, as well as the fibers of some cereals, triggers an intestinal dysbiosis with increased abundance in Enterobacteriaceae, Salmonella Escherichia/Shigella, and Clostridium XIVa, which, together with an aberrant immune response and a leaky gut, may trigger GID. It is observed that FS can be the product of previous GID. GID could provide information to generate a hypothesis of the bidirectional relationship between FS and GID. Emphasis is placed on the need for more studies with methodological rigor in selecting children with ASD, the need for homogeneous criteria in the evaluation of GID, and the adequate classification of FS in children with ASD.

## 1. Introduction

Autism spectrum disorder (ASD) is a heterogeneous and complex group of neurodevelopmental disorders. These are characterized by alterations in social interaction, cognitive functions, oral communication, social–emotional interaction, intellectual impairment, and motor and learning dysfunction, as well as self-restrictive obsessive–repetitive behaviors that manifest in early childhood and remain throughout an affected person’s life. The main manifestations of ASD include limitations in social communication and interaction, the inability to control or understand their own and others’ emotions, and alterations in proprioceptive and sensorial anomalies, such as the perception of pain and organic sensations (e.g., smell, taste, and sight). There are also limited interests in certain daily life activities, and a decreased intellectual capacity. It should be noted that the symptoms do not always present themselves in a linear and homogeneous chronology, and that their presentation varies according to the phenotype and severity of the ASD [1,2].

The worldwide prevalence of ASD is estimated at 1.13% (1:88 children), regardless of sex, race, socioeconomic status, or geographic determinants [2,3]. In addition to behavioral disturbances, several morbidities coexist with ASD, such as gastrointestinal disturbances/dysfunctions, which affect 8 out of 10 children with ASD [4]. The management of gastrointestinal symptoms (GIS) is relevant in the treatment of ASD because a relationship is observed between the progressive worsening of behavior and cognition with the severity of GIS [5]. The most frequently reported GIS in children with ASD are constipation, abdominal pain, and diarrhea, followed by chronic diarrhea, gastroesophageal reflux, nausea/vomiting, achalasia, and bloating. It should be noted that constipation has an important impact on the exacerbation or increase in behavioral and neurological symptoms [6,7].

Eating problems are common in children with ASD. Although most children have eating disorders in the first 2 years of life, those in children with ASD tend to be more severe, and persist throughout life. Food selectivity (FS) was hypothesized to play an important role in exacerbating GIS. FS, commonly referred to as picky/fussy eating, refers to a limited food repertoire in children [8]. This FS can be accompanied by a sensory aversion to food, characterized by a rejection of certain textures, temperature, flavors, colors, and smells; therefore, they tend to have preferences for a single type or brand of food, with unique characteristics of these sensory features. The diet of children with ASD who have FS is heterogeneous, and depends on the severity of the FS and concomitant GIS. When there is FS in ASD, the diet is restricted to the consumption of simple carbohydrates, and foods low in fiber and high in saturated fat. Children with ASD–FS prefer ultra-processed, calorie-dense foods, and food additives that impact their nutritional and gastrointestinal health status [9]. This review article aims to understand the possible relation between food selectivity and GIS in children with ASD.

## 2. Materials and Methods

The search for original articles focused on the electronic databases of PubMed, Google Scholar, and Elsevier. Publications in English from the last 15 years were included. The keywords “food selectivity”, “food neophobia”, “autism spectrum disorder”, “gastrointestinal symptoms”, “constipation”, “diarrhea”, and “oral over-responsiveness” were used. The previous search criteria were combined with other keywords separately: “feeding problems”, “mealtime problems”, food rejection, “gut microbiota”, “dysbiosis”, and “gastrointestinal disorders”. We included clinical trials, case studies, randomized controlled trials, and case series.

## 3. Eating Problems and Eating Behavior in Children with ASD

During early childhood, children are exposed to new experiences with food. These experiences are important sensory interactions that include new tastes, colors, temperatures, smells, and textures. Commonly, there is evidence of feeding problems in this period of new sensory experiences. More than a decade ago, it was reported that in neurotypical (NT) children (with normal development), there are about three times fewer feeding problems than in children with ASD (25–35% vs. 80% for NT and ASD, respectively) [10]. About 58% to 68% of parents or caregivers of children with ASD report problems with eating, such as selective food eating and food neophobia (Table 1) [11].

FS and feeding problems increase the risk of malnutrition, significant nutritional deficiencies, and GID in children with ASD (Table 1). In addition to the risks, it is reported that children with ASD have a 4–5 times higher risk of having eating disorders than NT children [19]. Parents and caregivers of children with ASD frequently report a concern regarding some of their eating behavior, such as very selective eating patterns restricted to only some food groups, even limiting themselves to consuming only four or five food types [20]. Unfortunately, these alterations in eating behavior usually persist throughout the lives of subjects with ASD [21]. The clinical study by Wallace et al. (2018) [22] reports a significant correlation in the severity of ASD and food neophobia with malnutrition risk, being overweight, and obesity, corroborated by BMI. This correlation depends on the foods included in the FS of children with ASD.

Food neophobias are expected at the beginning of exposure to foods other than milk in infants. Once complementary feeding begins, children are exposed to many sensory stimulations, flavors, colors, smells, and textures other than breast milk or infant formula. This incorporation leads to a sensory aversion of a physiological nature, characterized by food neophobia. This problem is usually transient in most NT children. However, it is reported that in children with ASD, most of these food neophobias persist [20].

Unfortunately, eating disorders in subjects with ASD do not seem to be transient, instead chronically affecting nutritional status. Metabolic alterations associated with modified intake patterns and eating behavior are common, with comorbidities associated with being overweight, obesity, and malnutrition the most common in subjects with ASD aged between 14 and 30 years. Without proper treatment, systemic problems related to malnutrition persist chronically [23].

The reality is that information about eating disorders in children with ASD is limited, even today. This limitation is mainly because these eating disorders do not represent a high risk to the health of subjects with ASD. Even with a diet restricted to several food groups, most children with ASD meet their caloric requirements at the expense of accepted (usually calorie-dense) foods [24,25].

### Food Neophobia in ASD

Although refusal of food and novel tastes is typical in NT children, they tend to disappear as the child is subjected to continuous exposure to tastes; however, children with ASD tend not to improve this acceptance over time. Food neophobia or “food rejection”, defined as a fear of trying new foods, is a restrictive eating behavior disorder [26]. Food neophobia is a key component of FS, generating a significant reluctance to try new foods. Children with high levels of food neophobia are more likely to avoid certain foods, and even avoid food groups altogether, negatively impacting their nutrient intake and diet quality.

Food rejection debuts in most children aged between two and six years. In NT children, this neophobia is transitory. From the sixth year, this eating behavior decreases while the child is more exposed to food, and the variety of foods in their diet increases. At this moment, the food begins to have a meaning beyond the nutritional state, involving the socio-emotional component of food. Regardless of the age at which children, and even adults, present food rejection, exposure to a greater variety of foods (even if they are not accepted, only exposed) reduces the severity of food rejection [27]. In addition to continuous exposure to a greater variety of foods, physical food transformation (such as bananas into banana ice cream, sweet potatoes into chips, carrot into juice, and kiwi into popsicles) also reduces the severity of food neophobia, and increases the consumption of fruits and vegetables [28].

In 2022, Marlow and Forestell, looking for factors associated with low consumption of fruits and vegetables in children with food rejection, found that children with parents who exhibit food neophobia of fruits and vegetables demonstrate food rejection of the same foods, even to a greater variety; that is, parents’ food-related behavior indirectly affects the eating behavior of their children [29]. The etiology of food rejection is still unknown; however, it is identified that some physiological, psychological, behavioral, and social components influence food neophobia development, duration, and severity [26].

Food rejection is extremely common in children with ASD, although its true prevalence is not reported [22,30]. Both conditions (ASD and food neophobia) have different characteristics and etiologies, for which they should be classified as different, not related, disorders. Studies report the prevalence of eating disorders or FS in children; however, food rejection is included, and not selectively. Besides, there are discrepancies in the concepts of food rejection because children are usually identified as “selective eaters”. More studies are needed to better classify food rejection, and other eating disorders, for the above.

Some studies focus on evaluating the role of sensory sensitivity in children with food rejection. The altered sensory sensitivities in the above-described ASD syndrome can negatively impact the interaction with food and the food environment [31].

## 4. Sensory Sensitivity in Children with ASD

It is common for individuals with ASD to have atypical sensory responses. These sensory alterations are a key point in clinical and diagnostic features, due to their high prevalence. Leekam et al. (2007) conducted a clinical study to evaluate sensory symptoms in children with ASD [32]. The study includes two groups of children with ASD, divided by functionality according to IQ (range IQ of 66–140: high-functioning children with ASD, *n* = 17; range IQ of 12–80: low-functioning children, *n* = 16), and one typically developing group (15 typical developing children with an IQ range of 81–138). The Diagnostic Interview for Social and Communication Disorders (DISCO) survey was conducted on the parents of children with ASD, to identify sensory disturbances in the study subjects. This first study reports that about 90% of individuals with ASD have alterations in sensory response compared to the typical development group children (33%). In addition, it is observed that IQ differences might impact sensory responses. High-functioning children with ASD show less atypical sensory responses than low-functioning children.

A meta-analysis by Ben-Sasson et al. (2009) reports a lower prevalence of impaired sensory response in individuals with ASD (45–46%) [33]. Unfortunately, these crucial differences in the estimated prevalence of sensory alterations may be due to the heterogeneity of the samples and subjectivity of the evaluation, since parents answer the surveys.

Children with ASD exhibit abnormal or atypical visual, tactile/oral, olfactory, and auditory sensory responses; the entire somatosensory system that provides information from the environment, and allows its interaction with it, presents alterations in the generation and processing of information [34]. This atypical sensory response to environmental stimuli makes the somatosensory pathway a stressor for ASD subjects. Several studies report a relationship between atypical sensory responses and various behavioral alterations in children with ASD [35]. Hence, sensory hyper- and hyposensitivity in the different sensory pathways can moderately or severely influence the ASD phenotype and its severity. In addition, those atypical sensory responses may persist throughout the life of people affected with ASD [36,37,38].

Sensory sensitivity strongly interferes with the daily activities of subjects with ASD [39]. For example, hyper-reactions to tactile stimuli, such as a hug or a particular scent, may result in rejection or a spontaneous negative response in the ASD subject. Furthermore, not only social interactions are affected by sensory sensitivity; sensory hypersensitivity is reported to be a significant component of feeding problems or disorders in children with ASDs [40]. Atypical sensory processing results in an over-response to food stimuli, perhaps leading to increased food rejection and food neophobia. The process of eating is quite complex, and includes the integration of various sensory aspects that influence individual acceptance and preference for food or its groups [41,42].

There are various characteristic patterns of ASD, but sensory sensitivity is significantly affected, leading to hyper or hypo-reactivity in these patients [43]. Their feeding struggles may be due to rituals at mealtime, chewing problems, vomiting, saturating themselves with food in the mouth, keeping food in the mouth for long periods without swallowing, food sensitivity, or neophobia [44]. Therefore, they tend to have preferences for a single type or brand of food, with unique characteristics such as texture, temperature, smell, color, or flavor [45]. The preferred texture is uniform or crunchy, with neutral colors, hot temperatures, and sweet flavors. The systematic review by Page et al. (2022) reports a positive correlation between FS and impaired sensory processing [46].

Children with ASD who exhibit sensory over-response tend to respond to stimuli with greater intensity, longer duration, and even faster than NT subjects or typical sensory responses. Some studies report a relationship between atypical oral sensory over-response and a rejection of the consumption of fruits and vegetables [21,47]. On the other hand, it is reported that a hyporesponsive sensory state can lead the child with ASD to a greater search for sweet, salty, or spicy foods to reach an adequate stimulus. Both processes make feeding difficult, and can lead to propitiating alterations in the feeding of children with ASD [48].

The clinical study by Dudova et al. (2011) identifies that children with ASD have a lower olfactory capacity than NT children, indicating a lower sensitivity and food identification in children with ASD [49]. Its identification is not established, nor is it very clear. However, research associates it with the severity of autism, the heterogeneity of populations, and screening tools such as clinical history or physical examination. For this reason, it is proposed that the research be carried out in the future, including large samples, autism severity classification, and homogeneous tools for its evaluation, in order to find statistically significant levels of association. Another clinical study, conducted by Sena et al. (2019), identifies that children with ASD have an olfactory dysfunction, particularly in the threshold olfactory capacity and odor identification [50]. Unfortunately, this study also has some critical limitations in terms of sample size (20 ASD children), although it does provide important insights into the nature of olfactory dysfunction in ASD.

Psychological studies by Bennetto et al. (2007) report a lower detection of sugars, and a greater preference for drinks such as juices [51]. Kral et al. (2015) conducted a clinical trial on adolescents with autism and NTs [52]. They indicate that adolescents with ASD show less precision in identifying sour and bitter tastes than NT adolescents. Regarding vision, children with ASD were presented with various images of different foods, showing a pattern of food stimuli compared to NT children. These children visualize complex dishes with greater preference and time than simple dishes: carrot vs. peas mix. However, children with ASD explored all dishes (complex and simple) similarly.

Children with ASD are more likely to reject food because of its texture and consistency, due to the atypical oral sensitivity they present, without forgetting smell and taste. Although visual and olfactory sensory sensitivities are reported to impact eating behaviors in children with ASD, they do not affect them as severely as tactile hypersensitivities, especially atypical oral responses. Oral defensiveness turns out to be a great trigger for neophobia and FS; oral over-response causes rejection of food textures. The study carried out by Muratori et al. (2015) identifies that children with ASD present a more severe food neophobia when they present an oral sensory over-response [53]. In addition, only a few studies evaluate the impact of oral sensory hyper-responsiveness in children with ASD. However, the authors agree that oral sensory hyper-responsiveness is the leading cause of food neophobia and FS in children with ASD.

It is not surprising that the foods least consumed by children with ASD are usually fruits, vegetables, legumes, meats, and some soft cereals. Generally, fruits and vegetables (in addition to their flavor and odors) are usually prepared in complex consistencies, which are generally rejected by children with ASD who present an oral sensory over-responsiveness. Therefore, children with ASD who exhibit this atypical sensory response often prefer crunchy, uniform, and semi-dry textures [45,47].

The etiology of oral sensory over-responsiveness in children with ASD is currently unknown. However, it is hypothesized that alterations in neurobiology and synaptic structure, characteristic of ASD, may significantly affect the transduction of these oral/tactile sensory stimuli. In addition, the assessment of oral sensory sensitivity also has several limitations. One of the main limitations is that the tools to assess this atypical sensory response are based on reports from parents and caregivers of children with ASD. These reports generate a necessary subjectivity that does not allow truthful conclusions to be made.

## 5. Food Selectivity “Peaking Eating/Fussy Eating” in Children with ASD

Currently, no diagnosis for FS exists, but is instead included within avoidant/restrictive food intake disorder (ARFID). ARFID was introduced into the DSM-V a little less than two decades ago; it was previously known as eating disorders in infancy and early childhood (0–6 years) [1]. This modification was because ARFID is not limited to age and duration [54]. However, not much is known about the sensory and behavioral profiles of children who have this disorder [55].

FS was included within the phenotypic characteristics of ASD since the primary description by Kanner (1968). FS is highly reported in children with developmental disorders, particularly in ASD, compared to NT children [56]. Although FS, eating problems, and rigid eating in children with ASD are highly reported, there is no standardized definition for FS. Bandini et al. (2010) propose a definition of FS based on the eating patterns of children with ASD [57]. This definition includes the classification of FS into three types: (1) food refusal, (2) limited food repertoire, and (3) high frequency of single food intake (Table 2). The refusal of food usually does not represent a risk to the child’s health with ASD because there are a certain number of foods that the child will consume. The severity of FS increases when there is a limited repertoire of foods because there are only single foods consumed for periods of 3 to 5 days. However, the high frequency of single food intake is limited to consuming a single food more than 4–5 times a day.

Sensory processes record, process, and organize the sensory information received by receptors in sensory organs, inducing an adequate response according to environmental demands. It is hypothesized that FS is directly related to the alterations in the somatosensory and proprioceptive pathways that are characteristic of ASD. These dysfunctions in sensory processing, due to alterations in connectivity and synaptic architecture, predispose children with ASD to present sensory hypersensitivities, especially oral sensory sensitivity [58]. The study conducted by Chistol et al. (2018) reports a direct association between oral sensory hypersensitivity with FS, being more common in children with ASD than in NT children [47]. Children with significant alterations in sensory pathways tend to have a more pronounced food neophobia, higher FS, and other food problems than children who have normal oral sensory capacity.

FS, and other eating disorders in children with ASD, is not only evident from a physiological perspective. From a psychological point of view, mealtime behaviors in children with ASD are peculiar. Studies report a higher number of mealtime rituals in children with ASD compared to NT children. These rituals range from observing food, arranging it by shape or color, and even playing with it or rejecting it, to requesting special utensils or packaging that determine their acceptance. Additionally, the psycho-emotional state contributes to FS and other eating problems [59].

It is common for subjects with ASD, or their parents and caregivers, to attribute FS to sensory aversions to smell (36%), taste (45%), texture (69%), temperature (22%), color, and appearance (58%), and even to the distribution of food. These problems suggest that the sensory component triggers this eating behavior. Unfortunately, sensory sensitivity, or aversion to food, restricts the intake of various foods. The above leads to eating behavior based on sensory preferences that includes only tolerable foods, or foods with manageable textures and organoleptic characteristics, which usually includes foods rich in starches, saturated fats, and dense in calories [60].

Unfortunately, caregivers or parents of children with ASD report that dietary selectivity is a bigger problem in the daily care of their children. Only a few studies evaluate the food intake and eating patterns of children with ASD, finding some critical evidence on food restrictions and preferences [20]. A study carried out in medical reports of 279 patients with ASD and a diagnosis of severe FS reports that at least 67% of children omit vegetables, and 27% omit fruits. Significant nutritional deficiencies, especially vitamin D, E, calcium, and fiber, are also reported in >50% of children with ASD. Regarding food preferences, children with ASD tend to prefer slices of bread made from refined wheat flour (70%), sweetened boxed cereals (50%), breaded chicken (chicken nuggets, 30–50%), sweetened yogurt (44%), crackers, and chips, among other fried and baked foods. Generally, the accepted foods are usually sweet, dry, and salty [9].

The development of metabolic disorders, and their associated comorbidities, is also one of the concerns for parents and caregivers of children with ASD. Unfortunately, both Molina-López et al. [61] and Sharp et al. (2018) [9] show associations between metabolic alterations, such as being overweight or underweight, with FS. Molina-López et al. report a higher presence of obesity and altered body composition (excess body fat) in children with ASD who present FS [61]. They also find that at least 50% of children with ASD have inadequate micronutrient intake compared to NT children.

In some cases, dietary selectivity can be so severe that it can lead to rare life-threatening nutritional deficiencies in children with ASD. The study conducted by Sharp et al. identifies at least twenty case reports involving 24 children diagnosed with ASD and severe FS (an acceptance of only 5–10 foods) that have scurvy [9]. Another case report involves the diagnosis of rickets with bilateral genu valgum. Rickets, considered a rare disease, is caused by a significant deficiency in vitamin D concentrations, causing a failure in bone mineralization during childhood. Similarly, the Hartman and Silver (2021) report concludes that this severe vitamin D deficiency is due to inadequate intake because of severe FS, associated with ASD [62].

## 6. Food Selectivity and Its Relationship with Gastrointestinal Disorders in ASD

ASD is associated with a long list of comorbidities, more common than in NT children. GID are one of the most frequent problems in children with ASD, with a three-fold increased risk of presenting any GIS, compared to NT children [7]. An NT child may manifest GI disturbances with signs and symptoms that are easy to identify clinically, such as vomiting, loss of appetite, abdominal pain manifested by shrinking of the abdominal region, or fetal positioning [63]. These GI symptoms in ASD are not presented in a typical way, and are usually identified by unusual behavioral changes, triggered spontaneously without any environmental stressor.

FS in ASD, depending on its severity, can lead to a significant nutritional imbalance [9,57,64,65], and be a direct contributing factor in the appearance of GIS. Children with ASD who exhibit severe degrees of FS, and additional sensory aversion to food, are reported to have inadequate intakes of protein, dietary fiber, and essential fatty acids, and a poor protein intake from any source [9]. This inadequate intake impacts the nutritional status of children with ASD, and directly affects GI physiology and intestinal microbiota balance, leading to GI alterations.

The evaluation and management of symptoms and GID are relevant in the treatment of ASD, because a close relationship is observed between the progressive worsening of behavior and cognition with the severity of GI symptoms [66]. Ferguson et al. (2019) find that children from 2 to 5 years old with ASD, compared to those from 6 to 18 years old, do not present behavioral and somatosensory alterations associated with gastrointestinal symptoms. Older children are anxious, avoid social contact, and have sensory hyperreactivity, constipation, and frequent abdominal pain [5].

The prevalence of GID is not established among ASD and NT people, due to its wide observed range between studies (9–91%) (Table 3). It is difficult to calculate the statistical significance of GID between ASD and NT children, but genetic heterogeneity, phenotypic manifestation, and clinical definition of GIS are just a few reasons [7]. However, Holingue et al. (2018) report that the mean prevalence of any GID in children with ASD is 46.8%. The most-reported GIS in the studies are constipation (22%) and diarrhea (13%) [67]. It should be noted that the study is not a systematic review; however, they report having carried out a similar methodology. Therefore, more analyses are needed to corroborate the prevalence of GI symptoms in subjects with ASD.

Another study investigates the prevalence of GI symptoms and dietary selectivity in a sample of children with ASD (*n* = 168; mean age: 43 months; range: 20–71 months). This study reports a prevalence of GI symptoms in 25.8% of children with ASD. To assess the impact of GI symptoms and behavioral disturbances, the Child Behavior Checklist (CBCL) survey was conducted. The behavioral patterns are observed with the Autism Diagnostic Observation Schedule (ADOS) assessment. Both behavioral assessment scales are applied to the children with ASD in the study (ASD GI-positive vs. ASD GI-negative). It is observed that children with ASD who present some GIS also present more behavioral alterations [16]. However, one of the most relevant limitations of this study continues to be the subjectivity of the assessment of GIS through parent surveys. The classification of the study group based on their age, and not on the severity or phenotype of the ASD, can also limit the presentation of GIS and their behavioral manifestations. The most frequent symptoms reported in children with ASD are constipation, abdominal pain, and diarrhea, followed by chronic diarrhea, gastroesophageal reflux, nausea/vomiting, achalasia, and bloating. It should be noted that constipation has an essential impact on the exacerbation of, or increase in, behavioral and neurological symptoms [6,7].

Eating behavior problems are closely related to the exacerbation of GIS. It is observed that children with ASD who present some pattern of FS tend to present higher levels of constipation than NT children; this may be explained by the fact that children with ASD who are picky eaters often avoid fruits and vegetables, and eat foods low in dietary fiber; these eating patterns promote constipation, even in NT subjects. A study by Ferguson in 2019 finds a positive association (r = 0.24, *p* = 0.030) between the appearance of GIS in the upper and lower digestive tract when dietary fiber intake is low in children with ASD. This same study finds that GIS associated with fiber intake also occurs when subjects consume a high amount of fiber (>600 g/fiber per month). This fiber intake in children with ASD is not typical, and is mainly due to an attempt to improve GIS with a high fiber intake, being counterproductive in most cases [5].

A recent study investigates the prevalence of GIS and FS in preschooler children with ASD to find a relationship between GIS and/or FS; however, only six GIS of the CBCL 11/2–5 scale of somatic symptoms are evaluated. In addition, the evaluation of FS is carried out through a single question (‘Your child eats well?’), in which parents specify how often and the type of food selectivity (they write the specification of their child’s eating behavior). They find a prevalence of GIS of 40.5%, with a high prevalence of children with FS (27.0%), and an important relationship between GIS and FS (12.27%) is found [17]. Unfortunately, this study does not analyze in-depth the degree of FS of subjects with ASD, nor the impact of FS on the presentation and severity of GIS. Besides, it presents several limitations in the assessment of GIS. In this study, all questionaries are applied to caregivers or parents of ASD subjects. No evaluation of food consumption and dietary patterns is carried out to identify the limitations in food repertoires and the eating behavior of the subjects with ASD vs. NTs. These limitations suggest the need for further studies in the evaluation of FS, and its impact on GIS severity.

It is common in studies that evaluate the impact of FS on the appearance and severity of GIS in children with ASD to base themselves on subjective questionnaires for the classification of FS and GIS. Therefore, their accuracy in the classification of FS, and the correlation with GIS, can be misleading. A recent study of 514 children (247 with ASD) reports a weak but significant correlation between GIS, FS, and mealtime problems with behavioral characteristics in the ASD group, but not with food supplement use [15]. Although the ASDs are diagnosed by an expert using validated instruments, such as the Autism Diagnostic Observation Schedule–Second Edition (ADOS–2) survey, the evaluation of the FS is conducted using an unvalidated instrument. This problem is also observed in the work of Harris et al. (2021), using the Generation R sample [66].

The pathophysiological mechanisms associated with GID in ASD are not fully understood. Although it is proposed that FS may be a consequence of GIS, there is no substantial evidence to support this theory. Some studies do not even find an association between FS and GIS in children with ASD. The study conducted by Postorino in 2015 does not find a relationship between GIS and FS in children with ASD, or NT children presenting with GIS. However, the sample size is 158 children with ASD, of whom only 13.9% have GIS (*n* = 22) [67].

For this reason, it is crucial to conduct studies with bigger sample sizes that allow us to see the effect of FS on GIS in children with ASD, and to include more variables that allow a broad view of the etiology of GID [68]. As the nature of these alterations is multifactorial, and includes genetical, morphophysiological, nutritional, psychological alterations, as well as sociodemographic and environmental factors, among others, if this list of variables is not considered, it can generate subjectivity in the results of the studies, hiding the true nature of the phenomena [69].

Some studies indicate that people with ASD present various structural, metabolic, and immunological alterations at a systemic level. These alterations seem to have an essential role in the common GI manifestations in some subjects with ASD and the ASD phenotype (Figure 1). Intestinal dysbiosis, permeability defects, various inflammatory processes, aberrant immune reactions [5,70], and other alterations in the microbiota–gut–brain axis are the main factors currently involved in the pathophysiology of ASD [71]. It should be noted that various metabolic, intestinal, and immunological abnormalities are present in large proportions of people with ASD, perhaps also associated with abnormalities in neuronal connectivity. However, this relationship is not corroborated today.

Although it is interesting that, among these common abnormalities in ASD, GIS are closely associated with neurological and behavioral symptoms, in both greater frequency and severity [4].

Some studies observe an association between gluten consumption and the exacerbation of GIS in people with ASD. Due to the higher prevalence of hypersensitivity to wheat proteins found in this population, nutritional interventions based on gluten-free diets are common. A subgroup of people with ASD and GIS that have immunological reactivity to wheat proteins are identified. High levels of IgG anti-gliadin antibodies (AGA), anti-gliadin deamidated peptides (DPG), and other gliadin-derived peptides are found in this group, such as 33-mer and peptides related to the CXCR3 receptor (e.g., QVLQQSTYQLLQELCCQGLW and QQQQQQQQQQQQILQQILQQ) [72]. This type of immune reaction is also described in people with non-celiac wheat sensitivity (NCWS). NCWS is a pathological entity produced by the ingestion of wheat. NCWS causes the exacerbation of intestinal and extraintestinal symptoms, which disappear once wheat, or the food that contains it, is eliminated from the diet. People with NCWS typically have IgA and IgG-AGA antibodies, with an inflammatory profile such as that seen in some people with ASD in response to wheat consumption, and whose gastrointestinal and behavioral discomfort is triggered by it. It is common for people with NCWS (but without ASD) to exhibit behavioral disturbances in response to wheat consumption, such as a foggy mind, headaches, depression, and sleep disturbances [73]. Unsurprisingly, Quan et al. [74] find a subgroup of people with ASD diagnosed with NCWS or celiac disease.

The relationship between FS and GIS has not yet been fully studied. Only a few studies search for physiological or molecular mechanisms associated with FS. As mentioned above, alterations in the gut microbiota may harm gut health in ASD subjects. It is known that food and eating patterns are some of the central modulators of the intestinal microbiota, and subjects with ASD, being selective eaters, affect the homeostasis of their intestinal microbiota. A recent study identifies a significant association between FS and the composition of the fecal microbiota, as well as an association between FS and GIS in children with ASD, unlike NT children (6.63 vs. 3.6 points in GI score analysis, respectively, *p* < 0.05) [75]. They find that children with FS (“picky eaters”) present more significant heterogeneity in the intestinal microbiota composition at the phyla level, where the genera Enterobacteriaceae, Salmonella Escherichia/Shigella, and Clostridium XIVa, are more abundant than in children with ASD without FS. These modifications in the intestinal microbiota seem to not be isolated in children with ASD, since NT children with FS also present this pattern in the intestinal microbiota. These data allow us to speculate that FS is a determining factor in changes in the intestinal microbiota, not only in children with ASD who are “picky eaters”. Table 3 shows a few studies that try to correlate FS with GIS.

Although many publications in the scientific literature point to an important relationship between alterations in the gut microbiota and the exacerbation of GID or GIS in people with ASD, the evidence is not conclusive, due to the limited number of studies, and the heterogeneity of their methods and criteria for selecting participants. A recent meta-analysis includes 16 original studies that attempt to relate the gut microbiota to GI symptoms in ASD subjects [74]. Three of the studies analyzed [75,76,77] find a relationship between gut microbiota abundance and GIS in children with ASD. Similar to the recent study mentioned above [75], the research carried out by Strati et al. (2017) finds a higher abundance of Escherichia/Shigella, and the cluster of clostridium XVIII associated with GIS and/or constipation (*p* < 0.05) [76]. On the other hand, the study by Luna in 2017 finds correlations between various gut microbiota bacteria and specific GI symptoms [77]. These studies provide valuable information on the role of the intestinal microbiota in the development of GIS/GID. However, due to their methodological difficulties, and the statistical differences between the studies, it is necessary to continue conducting studies before concluding. In addition, GIS/GID are usually multifactorial, and their etiology is not determined solely by alterations in a group of bacteria in the intestinal microbiota.

**Table 3 nutrients-14-02660-t003:** Most prevalent gastrointestinal (GI) symptoms, food selectivity (FS), and possible triggers in subjects with autism spectrum disorder (ASD).

Author	Sample Size (ASD/NT)	Prevalence of GIS	Most Prevalent GIS	Prevalence of FS	Possible Trigger of GIS	Study Type	Limitations
Ferguson et al., 2016 [78]	120 children with ASD (average age 11.8)		Constipation (42.5%) and low abdominal pain (9.2%).	FS not evaluated	Not discussed	The study is based on an indirect questionnaire, without directly assessing GIS or food intake by phone (QPGS Rome III questionnaire).	No food intake nor FS is evaluated; no eating behavior is evaluated, and the subjectivity of self-administered questionnaires.
Prosperi et al., 2017 [15]	163 preschoolers with ASD	28.5%	Constipation (22.1%) and low abdominal pain (7.4%)	27.0%	A relationship between GIS and FS (12.27%) is found	Study based on indirect questionnaire, without direct assessment of GIS or food intake.	No food intake is evaluated; FS is considered only in one item from CBCL 1 ½-5 score.
Ferguson et al., 2019 [5]	340 children with ASD (ages 2–18)	General prevalence not shown	Constipation (65%), stomachaches (47.9%), nausea (23.2%), and diarrhea (29.7%)	FS evaluated	Not discussed	Study based on indirect questionnaire, without direct assessment of GIS or food intake.	No FS is evaluated, no eating behavior is evaluated, and the subjectivity of self-administered questionnaires.
Babinska et al., 2020 [14]	247 subjects with ASD (2–17 years) vs. 267 controls (p 0.000)	88.7% of ASD subjects experienced GIS in the last 3 months, and 47.6% of ASD individuals present severe GIS	Constipation/hard stool consistency (61.9%), voluminous stools (51.0%), and bloating (49.4%)	High prevalence of FS (69.1%) compared to NT controls (37.1%), *p* = 0.000	FS and mealtime problems have a significant correlation with the severity of GIS. Children who exhibit FS have more GIS	Study based on indirect questionnaire, without direct assessment of GIS or food intake.	The sample is not randomly selected. No medical evaluation of GIS is performed, and the subjectivity of self-administered questionnaires
Tomova et al., 2020 [73]	46 children with ASD vs. 16 non-autistic children control	89.4% of ASD children vs. 87.5% of non-autistic children (*p* = 0.838) experience GI symptoms	Constipation (28.9%), bloating (35.6%), and abdominal pain (35.6%). Differences are observed only in constipation in ASD (*p* = 0.014)	57.7% of ASD children are “picky eaters” compared to controls (25%), *p* = 0.02.	FS modifies fecal microbiota composition. Children who exhibit FS have more GIS.	Study based on indirect questionnaire, without direct assessment of GIS. A food frequency questionnaire (FFQ) is used for dietary analysis.	A low number of participants, and subjectivity of self-administered questionnaires

QPGS Rome III: Questionnaire of Pediatric Gastrointestinal Symptoms (QPGS-III); CBCL 1 ½-5: Child Behavior Checklist for Ages 1(1/2)-5.

On the other hand, it is proposed that FS in ASD patients may occur because some of them present food rejection, gastroesophageal reflux, chronic constipation, delayed gastro emptying, and functional dyspepsia [78]. A retrospective case–cohort study partially confirms this proposal. In this study, a retrospective analysis of a sample of 45,286 patients with ASD and matched controls (*n* = 226,430) is carried out to associate the appearance of eosinophilic esophagitis in children with ASD and the presentation of alterations in eating behavior, mainly FS. This study finds a strong association between the diagnosis of FS and eosinophilic esophagitis [79]. Due to this, they propose that FS should not be considered a purely behavioral disorder, but rather associated with some physiological abnormality. However, as we have discussed above, few studies relate FS to the appearance of GI symptoms, or vice versa. Many studies focus mainly on assessing the effect of nutritional and eating behavior interventions to reduce FS and, in turn, GIS (Table 4). Unfortunately, the evidence continues to be inconsistent.

## 7. Conclusions

Unbalanced FS (low numbers of fruits and vegetables, or foods rich in fiber) seems to have an essential role in the appearance of GIS, due to diet’s predominant role in changes in the gut microbiota. Unbalanced FS produces disturbances in macro and micronutrients with a modulatory capacity of the gut microbiota. This disturbance promotes intestinal dysbiosis that, together with aberrant immune responses, alters gut permeability and favorable phenotypes in the gut microbiota of children with ASD, thereby triggering the appearance of GIS. This study’s findings suggest a relationship between food selectivity and GI symptoms in children diagnosed with ASD, but more studies are needed to prove it. It is suggested that the study of these problems be approached with standardized, validated, and specific methods, in order to avoid heterogeneity and information bias.

## Figures and Tables

**Figure 1 nutrients-14-02660-f001:**
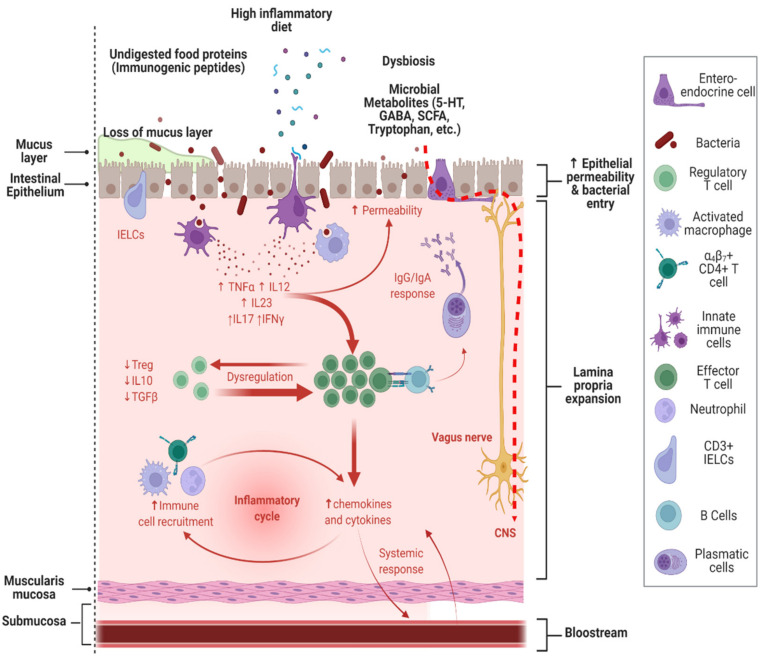
Alterations in the gut–immune–brain axis in ASD, and its role in GI alterations. Alterations in the intestinal microbiota produced by a restricted and inflammatory diet potentiate intestinal dysbiosis in subjects with ASD. The production of metabolites such as SCFAs, toxins, and neuroactive compounds, as well as immunogenic peptides derived from improper digestion, can cross the leaky gut, trigger an immune response, and alter brain functions. Immunogenic peptides and bacterial toxins can induce an inflammatory immune response, releasing cytokines into the systemic circulation and decreasing the tolerance response to dietary antigens. IELCs, intraepithelial lymphoid cells; TNF-α, tumoral necrosis factor-alpha; TGF-β, tumoral growing factor-beta; IFN-γ, interferon gamma; 5-HT, serotonin; GABA, γ-aminobutyric acid; SCFAs, short-chain fatty acids; CNS, central nervous system; IL, interleukin. Adapted from “Immune Response in IBD”, by BioRender.com. Retrieved from https://app.biorender.com/biorender-templates (accessed on 15 May 2022).

**Table 1 nutrients-14-02660-t001:** Most common eating problems in NT vs. ASD children.

Eating Problem	Prevalence ASD/NT (%)	Author	95% CI/*p*-Value
Food neophobia	58–67%/57.89%	Cherif et al., 2018 [11]	0.008
Pica	23.2%/8.4%	Fields et al., 2021 [12]	6.7 (5.1–8.8)
	11.8%/0%	Mayes and Zickgraf, 2019 [13]	NC
Food selectivity (fussy eating)	12.5%/NR	Inoue et al., 2021 [14]	0.778
	69.1%/37.1%	Babinska et al., 2020 [15]	0.0001
	27.0%/NC	Prosperi et al., 2017 [16]	N/R
	22.8%/3.5%	Cherif et al., 2018 [11]	0.008
Anorexia nervosa	22.9%/1%	Huke et al., 2013 [17]	NC
	23.65%/NC	Sedgewick et al., 2019 [18]	NC
	16.3%/NR	Inoue et al., 2021 [14]	0.778

NC, not compared vs. control; NR, not reported; NT, neurotypical.

**Table 2 nutrients-14-02660-t002:** Severity classifications in food selectivity (adapted from Bandini et al.) [57].

Classification	Definition	Questionary
Food refusal	There are a few foods that children with or without ASD will not consume, for preference or sensory reasons	Modified FFQ
Limited food repertoire	Foods consumed in 3 days, accepted for sensory reasons	3-day food diary, based on the NDSR
Severe food selectivity restricted to a single type of food	Foods consumed more than 5 times per day selectively	Modified FFQ

FFQ: food frequency questionnaire; NDSR: Nutrition Data System for Research.

**Table 4 nutrients-14-02660-t004:** Nutritional and behavioral clinical intervention studies in feeding problems.

Study	Total (N)	ASD (Group)	Age (Years)	Eating Problem	Intervention	Time (Weeks)	Control Group (TD)	Food Selectivity (95% CI)/Value *p*	Disruptive Mealtime Behaviors (95% CI)/Value *p*
Sharp et al., 2019 [80]	38	38	3–8	Moderate food selectivity	MEAL & PEP	16	NOT	−2.76 to −0.25	−6.16 to −0.69
Peterson et al., 2019 [81]	6	3	3–5	Mealtime behaviors	BAI	24	YES	N/A	0.001
Galpin et al., 2018 [82]	19	19	4–10	Feeding problems	SSN	12	NOT	0.001	0.13
Ghalichi et al., 2016 [83]	76	76	4–16	Stereotyped behaviors and social interaction	Gluten-free diet and regular diet	6	NOT	0.001	0.001
Thorsteinsdottir et al., 2021 [84]	81	33	8–12	Fussy eating	Taste education	7	YES	1.37 to 2.26	N/A
El-Meany et al., 2022 [85]	50	25	≥18	Feeding problems	Virgin coconut oil	12	YES	0.001	N/A
Santocchi et al., 2020 [86]	85	85	2–6	GI symptoms by food selectivity	DSF	20	YES	−0.68 to + 0.08	N/A
Johnson et al., 2015 [87]	14	14	2–7	Feeding problems	PT-F	16	NOT	0.05	N/A
Johnson et al., 2018 [88]	42	21	2–11	Feeding and mealtime problems	PT-F	20	YES	0.01	0.03
Gonzalez-Domenech et al., 2020 [89]	37	17	2–18	Behavior disorders	GFCF	24	YES	N/A	0.07
Kim et al., 2018 [90]	27	13	2–5	Food selectivity	Preventive program (exposure to vegetables)	24	YES	0.47	N/A

ASD: autism spectrum disorder, N/A: not application.

## Data Availability

Not applicable.

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
