# Peer review of "Food Selectivity and Its Implications Associated with Gastrointestinal Disorders in Children with Autism Spectrum Disorders"

_nutrients, 2022, doi:10.3390/nu14132660_

Round 1

Reviewer 1 Report

Very well written review.

Congratulations on this work

It is a comprehensive review that discuss carefully food selectivity and its association with GI disorders among children with autism.

Author Response

The paper is very interesting and actual. I suggest to speak about Non Celiac Gluten Sensitivity and its role in this disorder

R: Thanks. His suggestion was added in lines 437-454.

Reviewer 2 Report

The paper is very interesting and actual.  I suggest to speak about Non Celiac Gluten Sensitivity and its role in this disorder

Author Response

(The authors gave the same response as above.)

Reviewer 3 Report

The Authors described in a well-written and thorough narrative review, the possible association between food selectivity and autism. I read with interest the manuscript, but I have some comments.

Major comments:

-To my opinion more caution should be used. The manuscript is not a meta-analysis, “This review article is aimed to know the association between FS and GI symptoms in children with ASD” maybe should be replaces with “… aimed to describe the possible association…”. The same in the conclusion section. You did not analyze the data as in a systematic review and meta-analysis.

-from line 76 there is an important description of ASD that seem redundant. I suggest to move it in the introduction section, when you describe the disease.

-Table 1: to my opinion the p value of the differences should me added

-Table 2: “Foods consumed in 3 days *” what did you want say with “*”, something should be add in the notes.

Minor comments:

-line 62: typing error. Add "" in food rejection

-lines 133 reference is also need here, not only in the next phrase

-line 216: typing error, delete “)” after ASD

Author Response

-To my opinion more caution should be used. The manuscript is not a meta-analysis, “This review article is aimed to know the association between FS and GI symptoms in children with ASD” maybe should be replaces with “… aimed to describe the possible association…”. The same in the conclusion section. You did not analyze the data as in a systematic review and meta-analysis.

R: Thanks. His suggestion was followed throughout the manuscript.

-from line 76 there is an important description of ASD that seem redundant. I suggest to move it in the introduction section, when you describe the disease.

R: Thanks. It was moved to the introduction.

-Table 1: to my opinion the p value of the differences should me added.

R: Thanks. It was added

-Table 2: “Foods consumed in 3 days *” what did you want say with “*”, something should be add in the notes.

  1. Thanks. This typing error was deleted.

Minor comments:

-line 62: typing error. Add "" in food rejection

R: Thanks. It was added.

-lines 133 reference is also need here, not only in the next phrase

R: Thanks. It was added.

-line 216: typing error, delete “)” after ASD.

R: Thanks. It was deleted.
